# Using Word Embedding to Selectively Disclose Database Information

**Rajesh Bordawekar**
IBM T. J. Watson Research Center
Yorktown Heights, NY 10598, USA
bordaw@us.ibm.com

**Oded Shmueli** *
Computer Science Department
Technion, Haifa 32000, Israel
oshmu@cs.technion.ac.il

## Abstract

Database information may be disclosed in a variety of ways depending on the sensitivity of the stored information and the recipient's need to know. Traditionally, researchers have been concerned with preventing a recipient of the information from associating sensitive information (e.g., a disease) with specific individuals. However, other concerns may apply. For example, within an enterprise (Domingo-Ferrer et al., 2016), certain test results may be considered sensitive and should be only be openly disclosed to divisions concerned with these results. On the other hand, disclosing as much information as possible may also be in the enterprise's interest as it is not always clear what information may actually be useful to a division. We propose a mechanism that allows a discloser to exercise fine control over what is being disclosed and allowing disclosing information indirectly rather than directly. The mechanism is based on *word embedding*, a technique from Natural Language processing (NLP) in which each word is associated with a low dimensional (say, 200) vector of real numbers. These vectors are constructed so as to capture the meaning of the associated words. In disclosing vectors constructed based on sensitive information, rather than the information itself, we achieve *degrees of disclosures'*.

## 1 A Quick Introduction to Word Embedding

There are a few mechanisms for obtaining a vector representation of words in a language, called language, or word, embedding (Bengio et al., 2003). Such mechanismc use Neural Networks (NNs) (Bengio et al., 2003), log-linear classifiers (Mikolov et al., 2013a) and various matrix formulations (Levy & Goldberg, 2014).For example, a popular method is word2vec (Mikolov, 2013) that produces vectors that appear to capture syntactic as well semantic properties of words (Mikolov et al., 2013c;b). The exact mechanism employed by word2vec and suggestions for alternatives are the subject of current research (Goldberg & Levy, 2014; Pennington et al., 2014). However, word embedding may be applied to sequences other than natural language sentences, for example, the work of (Socher et al., 2013) explores capturing image semantics with word embedding.

## 2 Applying Word Embedding to Relational Databases

In (Bordawekar & Shmueli, 2017), a whole relation in a relational database may be converted into text (a process called textification), one tuple (record, row) at a time. As relational database columns are associated with a variety of data types (e.g., numeric,string, date, text) textification is a non-trivial process. However, once textification is done, each database token is associated with a vector capturing its *meaning*. This allows a whole new class of queries, called Cognitive intelligence (CI) queries that may be realized within standard SQL via user-defined functions (UDFs) (Bordawekar et al., 2017). CI queries operate both at the standard relational level as well as within a latent information level that exposes intra and inter-column hidden connections.

---

*Work done while the author was visiting IBM Research.

## 3 DEGREES OF INFORMATION DISCLOSURE

Consider a single relational database relation `R` with five columns: `A`, `B`, `C`, `D` and `E`. Further, assume its first column, `A`, contains the primary key, a string that is unique for each relation tuple (record, row). In disclosing R to a recipient we identify the following stages:

1. Deciding which columns should be completely eliminated, say due to a very high degree of sensitivity. In our example, we may decide to simply eliminate column `E`.

2. Deciding the content of which columns should be encrypted prior to producing word vectors. In our example, we may decide that column B be first encrypted. This keeps equality between equal entries in different tuples (rows) for *this* column, but severs identifying these values in other columns (inter-column severance) as well as hides the true nature of the content within en encrypted column.

3. Vector construction based on texitifying `R'` the modified relation `R`. This step associates a vector with each token in relation `R'`, see Figure 1.

4. Deciding which columns of R' are to be disclosed to the recipient. In our example, we may decide to disclose all columns, A, B, C and (the encrypted version of) D. The recipient will be presented with R" consisting of restricting R' to the disclosed columns.

5. Deciding which `R''` columns are to be disclosed to recipient should be first encrypted. Assume that column `B` is encrypted prior to disclosure in our example. The vector associated with an encrypted value is the one that was associated with the pre-encrypted value.

6. Disclosing pairs (w, v) where w is a token occurring in R" and v is the associated vector.

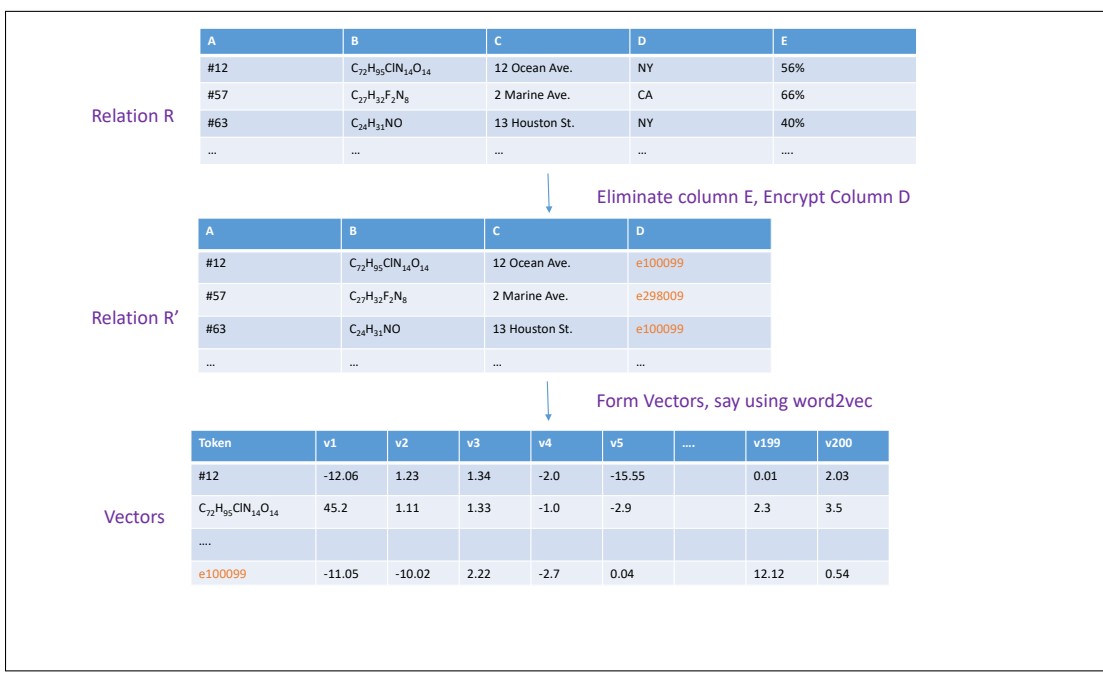

Figure 1: Producing word vectors from a modified relation.

The end result is that the recipient is presented with a relation (`R''` in our example) and with each token, its associated vectors, see Figure 2. The important point to note is that the recipient obtains significant additional information beyond the content of `R''`. The vectors, in fact, encode knowledge not present in `R''` i.e, knowledge accessible through the vectors, say using CI queries. On the other hand, *decoding* vectors and associating them to original relational tokens is a daunting task. *One can view a word embedding model as a one-way semantic hash from the source relational tokens to the meaning vectors.* This provides a level of information hiding that may be appropriate to many real-life situations. For example, suppose our table is describing employees, column `A`

is the employee badge number and column `C` records employee addresses (recall that column C is not encrypted in `R''`). While constructing vectors, the information in column C (suppose we also encrypt column `C` prior to releasing `R''`) is utilized in the vectors associated with the encrypted column C values. If two addresses are identical, this is easy to detect (although the addresses themselves cannot be discerned). If two addresses are close (say same town and street, different number) this information will be exposed in that their vectors are close (high cosine value). In this way, information may be hidden but exposed to a certain degree. This goes further, if one is interested in employees close to Joe Smith, this address information will also affect closeness of Joe Smith's column A vector to other employees column A vectors (along side other pieces of information in other columns).

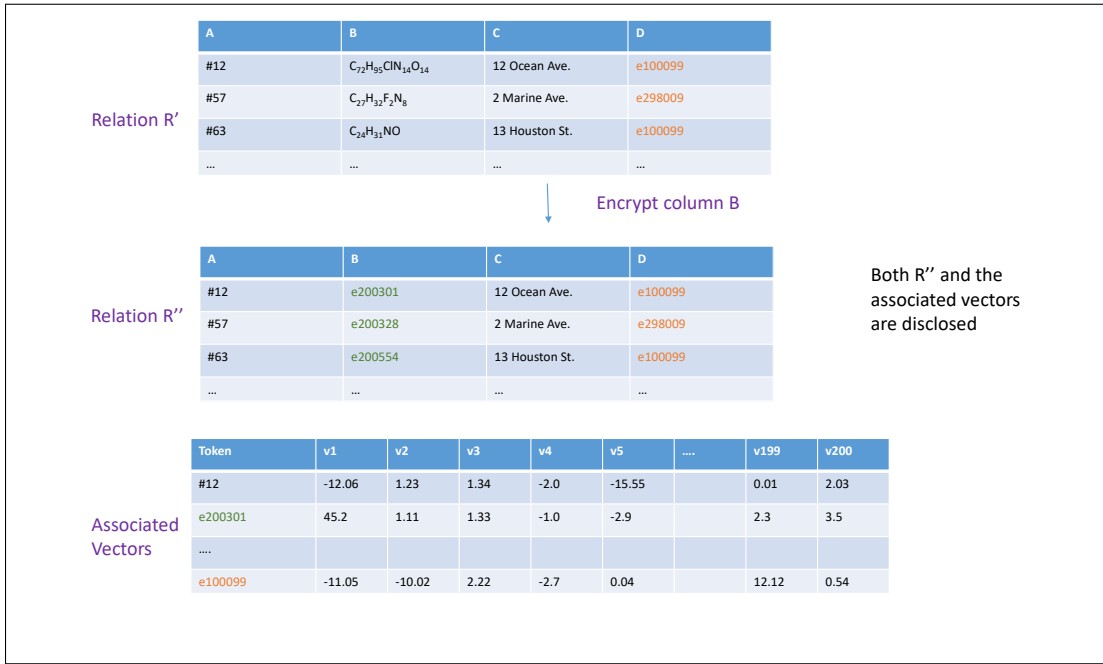

Figure 2: Producing word vectors from a modified relation.

## 4 DISCLOSING ADDITIONAL INFORMATION

The steps outlined above introduce increasing measures of information hiding: eliminating columns, encrypting prior to vector construction, eliminating a column when disclosing and encrypting disclosed columns. However, there are measures that reduce information hiding. One such measure that increases information exposure is the use of external sources, e.g., Wikipedia. During training, we can mix the text obtained by textifying the relation with text derived from external source(s). This way, the vectors of database tokens may encode closeness to terms that do not even appear in the database, thereby exposing additional information. For example, suppose that relation `R` deals with medical drugs. The word toxic may not appear in `R`. However, column `B` contains chemical formulae. Certain compounds may be identified by an external source as toxic. Training on both `R` and the external source may reveal closeness of a token of a column `B` (or `A`) vector. to the vector of toxic even though toxic does not appear in `R`.

## 5 CONCLUSIONS

Degrees of disclosures presents a wide range of possibilities for effective and measured disclosure. We plan to formally quantify these degrees and reason about their merits.

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
