# OpenReview forum: "USING WORD EMBEDDING TO SELECTIVELY DISCLOSE DATABASE INFORMATION"
_ICLR.cc/2018/Workshop — Reject_

### Official Review · AnonReviewer2 · 2018-03-09
**I sort of liked the idea of word embeddings as semantic hashes, but not sure there's a good fit here**

**Rating:** 3
**Confidence:** 3

**Review:**

Summary:

This paper suggest that encoding via word embeddings is a way to hide information in a database while still disclosing some information such as identity and similarity. (I could not speak how this relates to prior work in similar areas - outside my domain.)

Novelty:

This doesn't feel to me that novel, but the idea may not have been used in this domain.

Clarity:

Clear enough

Significance:

I doubt that this is highly significant

Quality:

This seems a fairly minor contribution.

It wasn't quite clear to me how you do textification for things like numeric fields, but maybe you effectively tokenize them and do it a digit at a time or something? There has been some recent work on "attacks" on neural models where effectively you are extracting source text from out of learned parameters. Could the approach here fall prey to the same idea and hence end up revealing more than the authors' suspect? There is no proof that the word vector encoding is really secure. See e.g., https://arxiv.org/pdf/1802.08232.pdf . The introduction paragraph felt a bit data and too elementary for the core ICLR audience.

Pros:

- This is sort of interesting as a perhaps novel use of word vectors
- It is sort of appealing to think of an embedding model as a one-way semantic hash.

Cons:

- There's just not a ton of content in this paper, even for a workshop paper. No experiments. No proofs/proof sketches. All there is is the idea of doing this and an illustrative example.

---

### Official Review · AnonReviewer1 · 2018-03-09
**Not good enough**

**Rating:** 2
**Confidence:** 5

**Review:**

The paper proposes a hierarchy of "information disclosure" for learning representations from databases, where the basic idea is to filter database entries depending on their sensitivity (e.g. encrypting privacy-sensitive database fields).

What is novel about this idea is altogether unclear to me; the paper is extremely sloppily written (see e.g. the mixed notation for R in the enumeration on page 2); and there is hardly any comparison to related work. There is a quick introduction to word embeddings, which is very out of place for ICLR (everyone will get the basic gist of word embeddings, hopefully), but no effort to relate the work to e.g. knowledge base representations, or representation learning over databases. See e.g. Nickel et al. (2016) "A Review of Relational Machine Learning for Knowledge Graphs", and references therein, e.g. Nickel et al. (2011) "A Three-Way Model for Collective Learning on Multi-Relational Data" and Bordes et al. (2011) "Learning Structured Embeddings of Knowledge Bases" for embedding structured information, or e.g. Seq2SQL for learning to interface with databases, or Key-Value Memory Networks for querying. Sections 1 and 2 generally feel like a throwback to 2014, while the field has moved on significantly.

The (rather laughably named) "Cognitive intelligence" queries should have been explained much better, and looking at that paper I also find it hard to see what is new about it from an ML perspective. This paper should not be accepted.

---

### Official Review · AnonReviewer3 · 2018-03-10
**Word Embeddings of partly encripted relational databases**

**Rating:** 4
**Confidence:** 4

**Review:**

The idea is simple. Encrypt some columns in relational databases and apply word2vec to obtain embedding vectors for the partly encrypted data, which generates embedding vectors of disclosed data end encrypted data.

I do not think this paper is suitable for ICLR. There is nothing new in representation learning methods and there is no evaluations about the effects of the partial encryption.  Is it possible to reach related  information using the generated embedding vectors even after partial encryption?

---

### Decision · Program_Chairs · 2018-03-20
**ICLR 2018 Workshop Acceptance Decision**

**Decision:**

Reject

**Comment:**

Based on the reviews, this paper has not been accepted for presentation at the ICLR workshop. However, the conversation and updates can continue to appear here on OpenReview.